# Grip Strength Correlates with Mental Health and Quality of Life after Critical Care: A Retrospective Study in a Post-Intensive Care Syndrome Clinic

**DOI:** 10.3390/jcm10143044

**Published:** 2021-07-08

**Authors:** Kensuke Nakamura, Ayako Kawasaki, Noriyo Suzuki, Sayaka Hosoi, Takahiro Fujita, Syohei Hachisu, Hidehiko Nakano, Hiromu Naraba, Masaki Mochizuki, Yuji Takahashi

**Affiliations:** Department of Emergency and Critical Care Medicine, Hitachi General Hospital, 2-1-1, Jonan-cho, Hitachi, Ibaraki 317-0077, Japan; ayako.kawasaki.os@hitachi.com (A.K.); noriyo.suzuki.zg@hitachi.com (N.S.); sayaka.hosoi.oq@hitachi.com (S.H.); arigato.jan.3rd.1439@gmail.com (T.F.); hachi084@gmail.com (S.H.); be.rann1988jp@gmail.com (H.N.); nrbhrm@gmail.com (H.N.); kurakan72@gmail.com (M.M.); yuji.mail@icloud.com (Y.T.)

**Keywords:** PICS, clinic, mental health, grip strength, HADS, critical care

## Abstract

Post-intensive care syndrome (PICS) is characterized by several prolonged symptoms after critical care, including physical and cognitive dysfunctions as well as mental illness. In clinical practice, the long-term follow-up of PICS is initiated after patients have been discharged from the intensive care unit, and one of the approaches used is a PICS clinic. Although physical dysfunction and mental illness often present in combination, they have not yet been examined in detail in PICS patients. Grip strength is a useful physical examination for PICS, and is reported to be associated with mental status in the elderly. We herein investigated the relationship between grip strength and the mental status using data from our PICS clinic. We primarily aimed to analyze the correlation between grip strength and the Hospital Anxiety and Depression Scale (HADS) score. We also analyzed the association between grip strength and the EuroQol 5 Dimension (EQ5D) score as quality of life (QOL). Subjects comprised 133 patients who visited the PICS clinic at one month after hospital discharge between August 2019 and December 2020. Total HADS scores were 7 (4, 13) and 10 (6, 16) (*p* = 0.029) and EQ5D scores were 0.96 (0.84, 1) and 0.77 (0.62, 0.89) (*p* ≤ 0.0001) in the no walking disability group and walking disability group, respectively. Grip strength negatively correlated with HADS and EQ5D scores. Correlation coefficients were r = −0.25 (*p* = 0.011) and r = −0.47 (*p* < 0.0001) for HADS and EQ5D scores, respectively. Grip strength was a useful evaluation that also reflected the mental status and QOL.

## 1. Introduction

Post-intensive care syndrome (PICS) is characterized by several prolonged symptoms after critical care, which mainly include physical and cognitive dysfunctions and mental illness [1]. Physical dysfunction is referred to as intensive care unit (ICU)-acquired weakness (ICU-AW) [2]. Since the body is damaged by severe conditions, the evaluation of and countermeasures against ICU-AW are frequently the focus of clinical practice. Mental illness also has a negative impact on patients with impaired quality of life (QOL) [3] and correlates with long-term survival after critical care [4]. Psychiatric conditions in the ICU, such as psychological stress [5], painful memories [6], and insomnia [7], independently affect the mental status.

In clinical practice, PICS requires a long-term follow-up after discharge [8]. One approach used is a PICS clinic [9], at which medical staff examine PICS after discharge from the ICU [10]. Although PICS clinics are actively operated in some countries, they are not common by global standards, particularly in Asian countries [11,12,13]. The first PICS clinic in Japan was opened in Hitachi General Hospital in 2019, and treats patients with severe conditions, including all ICU patients. While the Barthel index, Medical Research Council (MRC) score, and so on, are assessed for physical function, the Hospital Anxiety and Depression Scale (HADS) score for the mental status are recommended in evaluations of PICS in addition to other examinations [14]. Among several examinations, grip strength is one of the most useful physical examinations that may be easily assessed and detects slight muscle weakness as a continuous variable of muscle strength. Therefore, it is evaluated in all patients in our PICS clinic.

Although various PICS examinations conducted at PICS clinics revealed that physical dysfunction and mental illness often present in combination, they have not yet been investigated in detail in patients with PICS. A previous study reported that somatoform disorder was the most prominent symptom in a mental status evaluation after critical care [15]. Since PICS is considered to originate from damage to the body, physical dysfunction may be associated with mental illness. Grip strength is often coordinated with mental status, which is reported in the healthy individuals [16].

Therefore, we hypothesized that grip strength as a measure of physical performance correlates with the mental status in PICS. We herein investigated the relationship between grip strength and the mental status/QOL using outcome data from our PICS clinic.

## 2. Materials and Methods

This was a single-center retrospective study on PICS outcomes in our PICS clinic. Our PICS clinic was started in August 2019, and all ICU patients and patients with severe conditions admitted to Emergency and Critical Care Center were referred to the PICS clinic one month after hospital discharge. The PICS clinic opens every Thursday evening. Physicians, ICU nurses, and physiotherapists at the ICU participated in medical examinations. The physical, cognitive, and mental status of all patients were evaluated as described below. The present study was approved by the Ethics board of our hospital (2017-95).

Subjects comprised patients who visited the PICS clinic for the first time between August 2019 and December 2020. Our Emergency and Critical Care Center includes an 8-bed medical and surgical ICU with a 2:1 patient-nurse ratio for patients with severe conditions including postoperative acute deterioration, and a 10-bed emergency ward with a 4:1 patient-nurse ratio. Patients discharged from medical and surgical ICU and who had stayed in the emergency ward for more than or equal to 5 days were referred to the PICS clinic approximately one month after their discharge. An explanation of PICS with a written pamphlet was given to the patients and their families. Attendance to the PICS clinic was not mandatory. Only the first visit was examined in the present study, return visits were excluded.

We primarily aimed to analyze the correlation between grip strength and the HADS score. We also analyzed the association between grip strength and the EuroQol 5 Dimension (EQ5D) score as quality of life (QOL).

In our PICS clinic, physicians evaluated the physical status based on the presence of walking disability, muscle volume loss, and respiratory dysfunction, the mental status according to depression, anxiety, and sleep disorders, and the cognitive status based on memory impairment and executive function disorders. Walking disability was determined by whether the patient felt more difficulty in continuous walking for about 50 m on level surface, compared to the status before ICU admission. Physical performance was examined by physiotherapists using the following parameters: the Barthel index [17], the functional status score for ICU (FSS-ICU) [18], the MRC score [19], and grip strength (kg) in the left and right hands using a digital grip dynamometer (T2177, TOEI LIGHT, Saitama, Japan) and the mean of left and right strength. The proportion of grip strength ≥ Japanese age/gender matched control was also counted. HADS [20], Impact of Event Scale-Revised (IES-R) [21], and EQ5D scores [22] converted to Japanese QOL values (0–1) [23] were evaluated using a questionnaire as measures of the mental status, posttraumatic stress disorder and QOL, respectively. As a measure of cognitive function, the Mini-Mental State Examination (MMSE) [24] was used by nurses until January 2020, and the Short-Memory Questionnaire (SMQ) [25] from February 2020; however, some patients with deficits in cognitive functions could not be evaluated.

Regarding clinical information, age (years), the sequential organ failure assessment (SOFA) score, the acute physiology and chronic health evaluation (APACHEII) score at ICU admission, the length of hospital stay, the length of ICU stay, acute surgery and sepsis, and the induction of mechanical ventilation and renal replacement therapy, as well as their duration (days), were extracted. The length of ICU stay was calculated as the total stay in the ICU and emergency ward. Physical function assessment at hospital discharge was also extracted and analyzed.

## 3. Statistical Analysis

Continuous variables were expressed as means ± standard deviations and compared using the Student’s *t*-test when the null hypothesis was not rejected by the Shapiro-Wilk test. Continuous variables were expressed as medians (interquartile ranges) and compared using the Mann-Whitney U test when the null hypothesis was rejected by the Shapiro-Wilk test. Nonparametric paired values were compared with the Wilcoxon signed-rank sum test. Regarding categorical variables, the proportions of patients in the respective categories were calculated. Groups were then compared using the chi-squared test. To clarify the correlation between grip strength and the HADS/EQ5D scores, the Pearson correlation coefficient was calculated. All statistical analyses were conducted using JMP 14 software (SAS Institute Japan Inc. Tokyo, Japan). Results with a *p*-value < 0.05 are indicating with * as possessing a significant difference.

## 4. Results

The study outline is shown in Figure 1. A total of 2079 patients were admitted to the Emergency and Critical Care Center; 253 died in hospital and 1826 were discharged alive. Among these patients, 397 patients stayed in the medical and surgical ICU ≥ 1 day or in the emergency ward ≥ 5 days and PICS clinic reservations were made. In total, 133 patients visited the PICS clinic one month after hospital discharge and were analyzed in the present study.

Table 1 shows the baseline clinical data of patients: mean age, approximately 70 years; male, 67%; median SOFA score, 5 and APACHEII score, 15 at ICU admission; median length of hospital stay, 10 days; median length of ICU stay, 7 days; acute surgery, 34.6%; and sepsis, 43.6%. Outcomes, including examinations in the PICS clinic, are shown in Table 2. Physical function at hospital discharge was evaluated in 57 patients, because the other patients were discharged from the other departments. The following median values were obtained: the Barthel index, 85; FSS-ICU, 34; MRC score, 56; grip strength, 19.5 kg; and grip strength ≥ age/gender matched control, 0 (0%). The median values for these physical parameters in the PICS clinic were as follows: the Barthel index, 100; FSS-ICU, 35; MRC score, 58; grip strength, 20.75 kg; and grip strength ≥ age/gender matched control, 6 (4.5%). By the Wilcoxon signed-rank sum test in the 57 patients who were evaluated also at hospital discharge, there was significant improvements in Barthel index, FSS-ICU, MRC score and grip strength (*p* < 0.0001 * in all the parameters). Most patients had full scores for Barthel index, FSS-ICU, and the MRC score. On the basis of patient assessment by physicians, 60.2% exhibited walking disability, 52.6% muscle volume loss, indicating that minor physical dysfunction was not reflected in the Barthel index, FSS-ICU, or MRC score. 109 patients (82.0%) had at least one of the symptoms. Median HADS scores were 5 for depression, 3 for anxiety, and 8 in total. The median total IES-R score was 4 and the median EQ5D score was 0.84.

The baseline and outcomes were compared between with/without the walking disability determined by the physicians in Table 3. No significant differences were observed in age, sex, or disease severity. Patients in the walking disability group had a worse physical status at hospital discharge and in the PICS clinic. HADS and EQ5D scores were also higher in the walking disability group; total HADS scores were 7 (4, 13) and 10 (6, 16) (*p* = 0.029 *) and EQ5D scores were 0.96 (0.84, 1) and 0.77 (0.62, 0.89) (*p* ≤ 0.0001 *) in the no walking disability group and walking disability group, respectively. There was no correlation between physical parameters at PICS clinic and presence of muscle volume loss; Barthel Index was 100 (87.5, 100) and 100 (90, 100) (*p* = 0.62), FSS-ICU was 35 (35, 35) and 35 (34, 35) (*p* = 0.12), MRC score was 58.5 (56, 60) and 58 (54, 60) (*p* = 0.31), and grip strength was 21.6 (16.4, 29.1) and 20.8 (14.4, 28.1) (*p* = 0.65) in the no muscle volume loss group and muscle volume loss group, respectively.

The relationships between grip strength and HADS/EQ5D scores were also analyzed (Figure 2). Grip strength at hospital discharge (Figure 2a–c) and in the PICS clinic (Figure 2b,d) were negatively correlated with total HADS scores and EQ5D scores in the PICS clinic. Correlation coefficients were r = −0.25 (*p* = 0.011 *) and r = 0.47 (*p* < 0.0001 *) between grip strength and total HADS and EQ5D scores in the PICS clinic, respectively.

## 5. Discussion

Grip strength reflected minor physical weakness and was correlated with mental status and QOL. Mental status and QOL were correlated with physical function in the PICS clinic.

Although few studies have investigated the association between physical dysfunction and mental illness in PICS, such a relationship may not be observed in the acute phase of critical care. While physical dysfunction is generally the worst in the acute phase and improves gradually after ICU discharge [26], mental illness develops worse rather after hospital discharge [27]. However, the body damage in the acute phase of severe conditions could possibly contribute to the development of mental illness in the late phase [15]. A previous study demonstrated that physical restraint in the ICU was associated with mental illness [28]. Moreover, joint contracture may be a contributing factor to the mental status [29]. These associations between each other facets may be the essence of PICS as a long-term morbidity.

Grip strength was a useful measure in the PICS clinic and was correlated with mental status and QOL. This correlation has been reported in the elderly and patients with chronic diseases. As grip strength has been associated with anxiety [30] and depression [16] in the elderly, it has attracted attention in the evaluation of frailty. Grip strength is also correlated with the mental status of patients with chronic diseases, including middle-aged adults [31]. Therefore, mental illness may indirectly decrease grip strength. To the best of our knowledge, this is the first study to report this correlation in PICS after critical care. Upper body dysfunction may be severe in PICS [32], and may contribute to mental illness by decreasing the activities of daily living. Therefore, since an evaluation of upper body strength may be important in PICS, grip strength is a good marker of physical strength and may be assessed as a continuous variable, at least in the PICS clinic.

These investigations can be conducted via PICS clinic operation. There is currently no evidence to show that interventions at a PICS clinic prevent or ameliorate the symptoms of PICS [33]. However, ICU staff can hear the patients’ complaints and correspond to them at PICS clinic, and may select effective countermeasures for PICS via outcome feedback [10]. Although physical function may be the most strongly affected at hospital discharge, it gradually recovers in most cases [26], whereas the mental status frequently deteriorates with time after discharge [34,35]. Therefore, the evaluation of PICS during hospital admission may be inadequate, and an approach such as a PICS clinic appears to be necessary.

We should take care to interpret the study results in light of the fact that the most of patients in this study were relatively mild and moderate cases of PICS, especially for the physical facet. As the physical scores for activity of daily living were full in the most cases, patients with severe physical dysfunction may not have been able to come to the PICS clinic. This is the potential limitation of PICS clinic. However, since 82% of patients had some kind of the symptoms, many patients who visited the PICS clinic had suffered from PICS, and it would be meaningful to examine them, even if they were not serious.

The present study included several limitations. Only ICU patients who visited the PICS clinic were analyzed. We could not analyze the detailed reasons why the other patients did not visit the PICS clinic. Therefore, the results obtained need to be carefully interpreted for the total ICU population. PICS in all ICU patients in our hospital may have been worse than suggested by the present results. Furthermore, since this was a retrospective study, it was not possible to examine confounding factors for PICS. The causality of mental illness remains unclear. Moreover, this is a single-center study. PICS would be strongly influenced by patient population treated in their ICUs and by treatments including ICU cares. There might be an evaluation bias in our PICS clinic. Thus, single-center data cannot be directly applied to other hospitals. Large multicenter prospective studies are warranted in the future.

## 6. Conclusions

Grip strength was a useful evaluation that reflected the mental status and QOL in patients who visited the PICS clinic. The mental status and QOL was negatively correlated with grip strength.

## Figures and Tables

**Figure 1 jcm-10-03044-f001:**
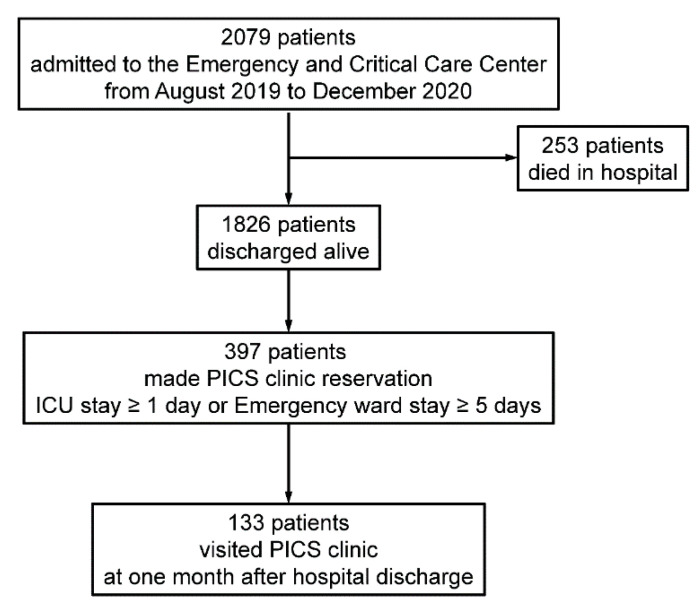
Study outline. ICU, intensive care unit; PICS, post intensive care syndrome.

**Figure 2 jcm-10-03044-f002:**
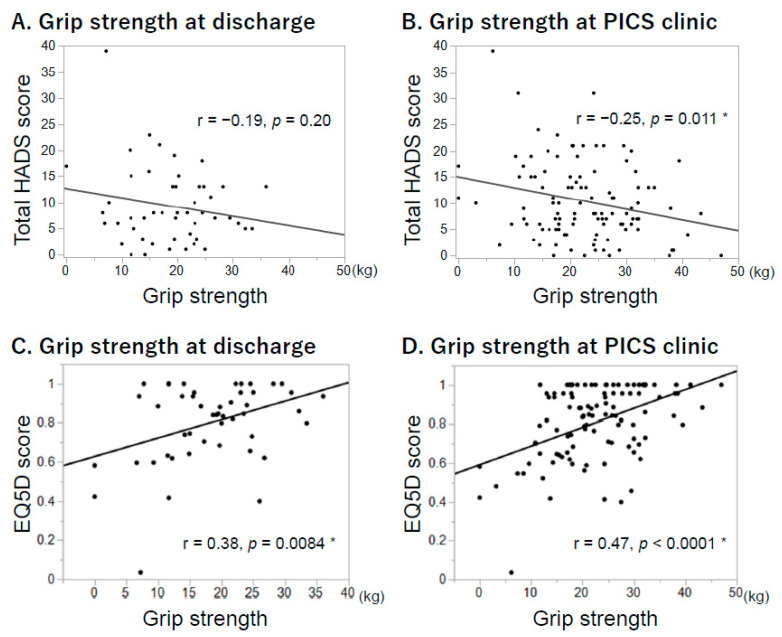
Relationships between grip strength and mental status/quality of life scores. All data and the relationships between grip strength and HADS/EQ5D scores are shown. HADS and EQ5D scores were evaluated in the PICS clinic. (**A**) Grip strength at discharge and HADS scores (*n* = 57). (**B**) Grip strength in the PICS clinic and HADS scores (*n* = 133). (**C**) Grip strength at discharge and EQ5D scores (*n* = 57). (**D**) Grip strength in the PICS clinic and EQ5D scores (*n* = 133). Results with a *p*-value < 0.05 are indicating with * as possessing a significant difference. HADS, Hospital Anxiety and Depression Scale; EQ5D, EuroQol 5 Dimension.

**Table 1 jcm-10-03044-t001:** Baseline characteristics on admission to the ICU.

*N*	133
age, years	69.8 ± 14.2
sex (male)	90 (67.6%)
SOFA	5 (4, 8)
APACHEII	15 (9, 18.75)
length of hospital stay	10 (4, 14)
length of ICU stay	7 (3, 12)
acute surgery	46 (34.6%)
sepsis	58 (43.6%)
mechanical ventilation, days	64 (48.1%), 2 (1, 7)
renal replacement therapy, days	25 (18.8%), 3 (1, 7)

SOFA, sequential organ failure assessment score; APACHEII, acute physiology and chronic health evaluation.

**Table 2 jcm-10-03044-t002:** PICS outcomes at hospital discharge and in the PICS clinic.

	*N*	133
At hospital discharge		
Physical status	Barthel index (*n* = 57)	85 (60, 95)
	FSS-ICU (*n* = 57)	34 (25, 35)
	MRC score (*n* = 57)	56 (48, 60)
	grip strength (kg) (*n* = 57)	19.5 (12, 25)
	grip strength ≥ age/gender matched control (*n* = 57)	0 (0%)
In the PICS clinic		
Physician assessment	At least one of the following symptoms	109 (82%)
	walking disability	80 (60.2%)
	muscle volume loss	70 (52.6%)
	respiratory dysfunction	29 (21.8%)
	Depression	17 (12.8%)
	Anxiety	20 (15%)
	sleep disorder	23 (17.3%)
	memory impairment	43 (32.3%)
	executive function disorders	27 (20.3%)
Physical status	Barthel index	100 (90, 100)
	FSS-ICU	35 (35, 35)
	MRC score	58 (54, 60)
	grip strength (kg)	20.75 (15.75, 28.25)
	grip strength ≥ age/gender matched control	6 (4.5%)
Mental status	total HADS	8 (5, 15)
	HADS (depression)	5 (3, 10)
	HADS (anxiety)	3 (1, 6)
	total IES-R	4 (1, 9)
	IES-R (Intrusion)	1 (0, 3)
	IES-R (Avoidance)	0 (0, 3)
	IES-R (Hyperarousal)	1 (0, 4)
Cognitive status	MMSE (*n* = 40)	27.5 (23, 30)
	SMQ (*n* = 24)	36 (26.5, 41)
Quality of life	EQ5D	0.84 (0.68, 0.96)

FSS-ICU, functional status score for the ICU; MRC, Medical Research Council score; HADS, Hospital Anxiety and Depression Scale; IES-R, Impact of Event Scale-Revised; MMSE, Mini-Mental State Examination; SMQ, Short-Memory Questionnaire; EQ5D, EuroQol 5 Dimension.

**Table 3 jcm-10-03044-t003:** PICS outcomes with/without walking disability in the PICS clinic.

		**No Walking Disability**	**Walking Disability**	***p*** **-Value**
	*n*	53	80	value
Admission data	age	72.2 ± 11.9	68.2 ± 15.7	0.14
	sex (male)	39 (73.5%)	52 (65%)	0.30
	SOFA	5 (4, 8)	5 (4, 9)	0.94
	APACHEII	16 (11.5, 19.5)	15 (9, 18.5)	0.40
	length of hospital stay	7 (3, 13)	7 (3, 14)	0.94
	length of ICU stay	6 (3, 11)	7 (3, 13)	0.59
At discharge	*n* (%)	19 (35.8%)	38 (47.5%)	0.50
Physical status	Barthel index at discharge	90 (70, 100)	80 (48.75, 95)	0.0056 *
	FSS-ICU at discharge	35 (28, 35)	34 (25, 35)	0.086
	MRC score at discharge	58 (56, 60)	55 (48, 58.25)	0.0010 *
	grip strength (kg) at discharge	19.5 (15.5, 31)	19.0 (11.65, 23.65)	0.17
	grip strength ≥ age/gender matched control	0 (0%)	0 (0%)	1
At PICS clinic				
Physical status	Barthel index	100 (100, 100)	100 (80, 100)	0.012 *
	FSS-ICU	35 (35, 35)	35 (34, 35)	0.0025 *
	MRC score	60 (58, 60)	58 (53, 60)	0.0010 *
	grip strength (kg)	24.75 (18.20, 31.30)	20.10 (13.10, 26.35)	0.0021 *
	grip strength ≥ age/gender matched control	5 (10.9%)	1 (1.3%)	0.015 *
Mental status	total HADS	7 (4, 13)	10 (6, 16)	0.029 *
	HADS (depression)	4 (1, 7)	7 (4, 10)	0.013 *
	HADS (anxiety)	2 (1, 4)	4 (1, 7)	0.046 *
	total IES-R	3 (1, 8)	4.5 (1, 10)	0.25
	IES-R (Intrusion)	1 (1, 2.75)	2 (0, 4)	0.43
	IES-R (Avoidance)	0 (0, 2)	0 (0, 3)	0.49
	IES-R (Hyperarousal)	1 (0, 3)	1 (0, 5)	0.26
Quality of life	EQ5D	0.96 (0.84, 1)	0.77 (0.62, 0.89)	<0.0001 *

Results with a *p*-value < 0.05 are indicating with * as possessing a significant difference. SOFA, sequential organ failure assessment score; APACHEII, acute physiology and chronic health evaluation; FSS-ICU, functional status score for the ICU; MRC, Medical Research Council score; HADS, Hospital Anxiety and Depression Scale; IES-R, Impact of Event Scale-Revised; MMSE, Mini-Mental State Examination; SMQ, Short-Memory Questionnaire; EQ5D, EuroQol 5 Dimension.

## Data Availability

The datasets generated and analyzed during the present study are available from the corre-sponding author upon reasonable request.

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
