# Peer review of "Grip Strength Correlates with Mental Health and Quality of Life after Critical Care: A Retrospective Study in a Post-Intensive Care Syndrome Clinic"

_jcm, 2021, doi:10.3390/jcm10143044_

Round 1

Reviewer 1 Report

This study retrospectively analyzed post-ICU clinic data and reported that there were correlations between grip strength, mental health, and QoL. I think the topic of the study is interesting and timely. However, considering that the concept of PICS itself is multifaceted (physical, mental, and cognitive) and that the three domains are related to each other, the study findings do not appear to be new. My comments to help improve the quality of the manuscript are as follows:

  1. The authors described the purpose of the study as "investigated the relationship between physical dysfunction and the mental status/QOL", and physical dysfunction and grip strength appear to be different. Please describe consistently throughout the manuscript (including title to conclusions) which of the two was the main variable to this study.
  2. Please describe in more detail the measurement of "physical dysfunction".
  3. Are all variables in the post ICU clinic data measured at the same time? Please clearly state at what point after discharge.
  4. How did you calculate your EQ scores? Most studies report the index value of EQ (0-1) or VAS(0-100).
  5. Some of the values in Table 3 do not appear to differ between groups. For example, even though the FSS-ICU values are equal to 35 in both groups, it is described as having a statistically significant difference, so please check.
  6. Most of the clinic data appear to fall within the normal range. Please add a discussion about the degree of PICS of the study participants. 

Author Response

Reviewer 1

This study retrospectively analyzed post-ICU clinic data and reported that there were correlations between grip strength, mental health, and QoL. I think the topic of the study is interesting and timely. However, considering that the concept of PICS itself is multifaceted (physical, mental, and cognitive) and that the three domains are related to each other, the study findings do not appear to be new. My comments to help improve the quality of the manuscript are as follows:

Our answer: We would like to thank the reviewer for his/her review and meaningful comments on our manuscript. We revised it as the reviewer suggested.

  1. The authors described the purpose of the study as "investigated the relationship between physical dysfunction and the mental status/QOL", and physical dysfunction and grip strength appear to be different. Please describe consistently throughout the manuscript (including title to conclusions) which of the two was the main variable to this study.

Our answer: Thank you for these important comments. Our main purpose/conclusion was the relationship between grip strength and mental status/QOL in the PICS clinic. Therefore, we revised it throughout the manuscript including abstract, with a careful mention that main objective of the study is the relationship between grip strength and mental status/QOL.

  1. Please describe in more detail the measurement of "physical dysfunction".

Our answer: Thank you for this important comment and explanation was lacking. The physicians determined it by medical interview of walking disability, in which the patient feels more difficulty in a consecutive walking for about 50 meters on level surface, comparing with the status before ICU admission. We consider that we should not say it as "physical dysfunction", therefore, we explained it as “walking disability” throughout the manuscript and added the determination in method section as follows.

“Walking disability was determined whether the patient felt more difficulty in a consecutive walking for about 50 meters on level surface, comparing with the status before ICU admission.”

  1. Are all variables in the post ICU clinic data measured at the same time? Please clearly state at what point after discharge.

Our answer: We apologize for the lack of information. Yes, all the patients visited our PICS clinic one month after hospital discharge. We added this information in the abstract, method and result sections as follows.

“Subjects comprised 133 patients who visited the PICS clinic at one month after hospital discharge between August 2019 and December 2020.”

“all ICU patients and patients with severe conditions admitted to Emergency and Critical Care Center were referred to the PICS clinic one month after hospital discharge.”

“In total, 133 patients visited the PICS clinic one month after hospital discharge and were analyzed in the present study.”

  1. How did you calculate your EQ scores? Most studies report the index value of EQ (0-1) or VAS(0-100).

Our answer: We apologize, we calculated simple total EQ5D scores in the previous manuscript. In the revised one, we calculated EQ5D converted to Japanese QOL values (0-1). We revised the tables and figure with EQ5D and the sentences in method and results sections as follows.  

“Impact of Event Scale-Revised (IES-R)[21], and EQ5D scores [22] converted to Japanese QOL values (0-1) [23] were evaluated using a questionnaire as measures of the mental status, posttraumatic stress disorder, and QOL, respectively.”

“The median total IES-R score was 4 and the median EQ5D score was 0.84.”

“EQ5D scores were 0.96 (0.84, 1) and 0.77 (0.62, 0.89) (p=<0.0001*) in the no walking disability group and walking disability group, respectively.”

  1. Some of the values in Table 3 do not appear to differ between groups. For example, even though the FSS-ICU values are equal to 35 in both groups, it is described as having a statistically significant difference, so please check.

Our answer: Thank you for this comment. I confirmed some of the values were wrong in table.3 and revised it to the right values. Barthel index, FSS-ICU and MRC score at PICS clinic were similar in median values, however, we confirmed the p values were right in the previous table.3 and there were significant differences by Wilcoxon test. For example of FSS-ICU at PICS clinic, median and 25% quartile were similar in both group, however, 10% quartile was 34.7 vs 6.4, and mean was 34 ± 4.7 vs 30.5 ± 9.9,  respectively. 

  1. Most of the clinic data appear to fall within the normal range. Please add a discussion about the degree of PICS of the study participants.

Our answer: Thank you for this suggestion. We agree that the most of patients in this study were relatively mild and moderate cases of PICS, especially for physical facet. However, 82% of patients had some kind of the symptoms, and we could not say that they were not PICS because of non-severity. We added these analysis and discussion in the results and discussion sections as follows.

“109 patients (82.0%) had at least one of the symptoms.”

“We should take care to interpret the study results in the view that the most of patients in this study were relatively mild and moderate cases of PICS, especially for physical facet. As the physical scores for activity of daily living were full in the most cases, patients with severe physical dysfunction may not have been able to come to the PICS clinic. It is the potential limitation of PICS clinic. However, since 82% of patients had some kind of the symptoms, many patients who visited the PICS clinic had suffered from PICS and it would be meaningful to examine them, even if they were not serious.”

Reviewer 2 Report

Dear Authors,

thank you for the opportunity to review your manuscript showing that hand grip strength in a PICS clinic correlates with quality of life and mental status in ICU survivors.

General Comments

  • Please eliminate abbreviations from the title in order to enhance understandability.
  • Please include the study design in the title.

Introduction

  • The part regarding early mobilization does not add to the introduction but rather steers the reader away from the purpose of this investigation.

Methods

  • Please clearly state your primary endpoint. Is it the HADS score or the correlation to strength? If it is the correlation, do you have two primary endpoints? How did you account for that statistically?
  • Please include the correlation in the statistic section.
  • Please include the parameters and cut-off values that lead to determination of physical dysfunction.
  • How did you diagnose PICS? How many of your patients were suffering from PICS?

Results

  • Did muscle volume and strength correlate?
  • Hand grip strength should also be shown as percentage of age and gender-matched controls e.g. according to Dodds et al..
  • What were the reasons patients did not attend the PICS clinic after reservation?
  • Why were only 49 patients assessed at hospital discharge?
  • Do you observe improvement between hospital discharge and PICS clinic visit in the subset of 49 patients as it appears you compared these 49 to all 133? What statistical test was used for this comparison? Please include the p-values.
  • Please show the distribution of the 49 patients assessed at hospital discharge between no physical dysfunction and physical dysfunction in table 3.

Discussion

  • Has this association been shown during the acute phase of critical care?
  • Please give a more detailed insight on how this furthers knowledge of PICS.

Figures

  • Please correct all typing errors in Figure 1.
  • It is unclear if 2079 were discharged or if 1826 were discharged? At what point did the 253 patients die?

Author Response

Reviewer 2

Dear Authors,

thank you for the opportunity to review your manuscript showing that hand grip strength in a PICS clinic correlates with quality of life and mental status in ICU survivors.

Our answer: First of all, we would like to thank the reviewer for his/her careful review and meaningful comments on our manuscript. We revised it as the reviewer suggested.

General Comments

Please eliminate abbreviations from the title in order to enhance understandability.

Please include the study design in the title.

Our answer: We revised the title as follows.

“Grip strength correlates with mental health and quality of life after critical care: a retrospective study in a post-intensive care syndrome clinic”

Introduction

The part regarding early mobilization does not add to the introduction but rather steers the reader away from the purpose of this investigation.

Our answer: We agree with the reviewer’s comment and removed the sentences related to early mobilization from introduction and discussion section.

Methods

Please clearly state your primary endpoint. Is it the HADS score or the correlation to strength? If it is the correlation, do you have two primary endpoints?

Our answer: The primary aim of this study was to clarify the relationship between grip strength and mental status/QOL in the PICS clinic. As the other reviewer also suggested, we revised to state the primary aim throughout the manuscript including abstract. We also revised the sentences related to primary endpoint in abstract and method section as follows.

“We primarily aimed to analyze the correlation between grip strength and the Hospital Anxiety and Depression Scale (HADS) score. We also analyzed the association between grip strength and the EuroQol 5 Dimension (EQ5D) score as quality of life (QOL).”

How did you account for that statistically?

Please include the correlation in the statistic section.

Our answer: We used the Pearson correlation coefficient for analysis of the correlation between grip strength and the HADS/EQ5D scores. We added explanation in the statistic section as follows.

“To clarify the correlation between grip strength and the HADS/EQ5D scores, the Pearson correlation coefficient was calculated.”

Please include the parameters and cut-off values that lead to determination of physical dysfunction.

Our answer: Thank you for this comment. Our explanation was lacking. The physicians determined it by medical interview of walking disability, in which the patient feels more difficulty in a consecutive walking for about 50 meters on level surface, comparing with the status before ICU admission. Furthermore, we consider that we should not say it as "physical dysfunction", therefore, we explained it as “walking disability” throughout the manuscript and added the determination in method section as follows.

“Walking disability was determined whether the patient felt more difficulty in a consecutive walking for about 50 meters on level surface, comparing with the status before ICU admission.”

How did you diagnose PICS? How many of your patients were suffering from PICS?

Our answer: Thank you for this important comment. As we should not clearly say “the one is the PICS” or “the other is not” in the concept of PICS, there has been no clinical criteria for PICS and we have to treat patients as a broad definition of PICS even if they suffer from slight symptoms. Therefore, we added the patient proportion who suffered from at least one of the symptoms. 109 patients (82.0%) had at least one of the symptoms in our PICS clinic. We added this analysis in the results sections and table.2 as follows.

“109 patients (82.0%) had at least one of the symptoms.”

Results

Did muscle volume and strength correlate?

Our answer: Thank you for this comment, We analyzed it, however, there was no significant difference in physical parameters at PICS clinic between no muscle volume loss group and muscle volume loss group. We added these analyses in results section as follows.

“There was no correlation between physical parameters at PICS clinic and presence of muscle volume loss; Barthel Index was 100 (87.5, 100) and 100 (90, 100) (p=0.62), FSS-ICU was 35 (35, 35) and 35 (34, 35) (p=0.12), MRC score was 58.5 (56, 60) and 58 (54, 60) (p=0.31), and grip strength was 21.6 (16.4, 29.1) and 20.8 (14.4, 28.1) (p=0.65), in no muscle volume loss group and muscle volume loss group, respectively.”

Hand grip strength should also be shown as percentage of age and gender-matched controls e.g. according to Dodds et al..

Our answer: We added the proportion of grip strength ≥ Japanese age/gender matched control. There were few patients who had the age/gender matched grip strength.

What were the reasons patients did not attend the PICS clinic after reservation?

Our answer: Thank you for this comment. Although we could not analyze the detailed reasons why the other patients did not visit the PICS clinic, some of them might have more severe physical dysfunction and might not be able to come to the PICS clinic. We added these discussions in discussion and limitation section as follows.

“We should take care to interpret the study results in the view that the most of patients in this study were relatively mild and moderate cases of PICS, especially for physical facet. As the physical scores for activity of daily living were full in the most cases, patients with severe physical dysfunction may not have been able to come to the PICS clinic. It is the potential limitation of PICS clinic. However, since 82% of patients had some kind of the symptoms, many patients who visited the PICS clinic had suffered from PICS and it would be meaningful to examine them, even if they were not serious.”

“We could not analyze the detailed reasons why the other patients did not visit the PICS clinic.”

Why were only 49 patients assessed at hospital discharge?

Our answer: At first, we apologize that the number of patients 49 in the previous manuscript was wrong and 57 was right. We could not evaluate the other patients, because they were transferred from ICU to the other departments, and discharged from the other departments. We added these explanations in results section as follows.

“Physical function at hospital discharge was evaluated in 57 patients, because the other patients were discharged from the other departments.”

Do you observe improvement between hospital discharge and PICS clinic visit in the subset of 49 patients as it appears you compared these 49 to all 133? What statistical test was used for this comparison? Please include the p-values.

Our answer: Thank you for this comment. We should compare them in 57 patients to 57 patients with paired statistics. We analyzed it with the Wilcoxon signed-rank sum test and there was significant improvements in all the physical parameters. We added the statistics in method section and revised the related sentence in result section as follows.

“Nonparametric paired values were compared with the Wilcoxon signed-rank sum test.”

“By the Wilcoxon signed-rank sum test in the 57 patients who were evaluated also at hospital discharge, there was significant improvements in Barthel index, FSS-ICU, MRC score and grip strength (p<0.0001* in all the parameters).”

Please show the distribution of the 49 patients assessed at hospital discharge between no physical dysfunction and physical dysfunction in table 3.

Our answer: We added the distribution of the 57 patients assessed at hospital discharge in table.3.

Discussion

Has this association been shown during the acute phase of critical care?

Please give a more detailed insight on how this furthers knowledge of PICS.

Our answer: Thank you for this meaningful comment. Such a relationship would not be observed in the acute phase of critical care, because physical dysfunctions is obviously predominant, and mental illness or QOL decline are indistinctive in the acute phase. We added these discussion in the discussion section as follows.

“Although few studies have investigated the association between physical dysfunction and mental illness in PICS, such a relationship may not be observed in the acute phase of critical care. While physical dysfunction is generally the worst in the acute phase and improves gradually after ICU discharge [26], mental illness develops worse rather after hospital discharge [27]. However, the body damages in the acute phase of severe conditions possibly contribute to the development of mental illness in the late phase [15]. A previous study demonstrated that physical restraint in the ICU was associated with mental illness [28]. Moreover, joint contracture may be a contributing factor to the mental status [29]. These association between each other facets may be the essence of PICS as the long-term morbidity.”

Figures

Please correct all typing errors in Figure 1.

It is unclear if 2079 were discharged or if 1826 were discharged? At what point did the 253 patients die?

Our answer: We revised figure.1 with the right information.

Reviewer 3 Report

We thank authors for this interesting paper on PICS evaluation at a PICS clinics. The English is correct. Ethics are precised.  Methods and statistical analysis are adapted to the question.In limitation part, i think it is important to precise the monocentric aspect of the study

Author Response

Reviewer 3

We thank authors for this interesting paper on PICS evaluation at a PICS clinics. The English is correct. Ethics are precised. Methods and statistical analysis are adapted to the question.In limitation part, i think it is important to precise the monocentric aspect of the study.

Our answer: We would like to thank the reviewer for his/her review of our manuscript. We agree with our limitation of a single center study. We revised the limitation section as follows.

“Moreover, this is a single center study. PICS would be strongly influenced by patient population treated in their ICUs and by treatments including ICU cares. There might be an evaluation bias in our PICS clinic. Thus, single center data cannot be directly applied to other hospitals. Large multicenter prospective studies are warranted in the future.”

Round 2

Reviewer 2 Report

Thanks for your revisions -the paper benefited from these alot. I would consider a final check for English style and grammar by a native speaker.